# Efficient Cardiac Differentiation of Human Amniotic Fluid-Derived Stem Cells into Induced Pluripotent Stem Cells and Their Potential Immune Privilege

**DOI:** 10.3390/ijms21072359

**Published:** 2020-03-29

**Authors:** Yi-Hsien Fang, Saprina P.H. Wang, Zi-Han Gao, Sheng-Nan Wu, Hsien-Yuan Chang, Pei-Jung Yang, Ping-Yen Liu, Yen-Wen Liu

**Affiliations:** 1Institute of Clinical Medicine, College of Medicine, National Cheng Kung University, Tainan 70457, Taiwan; eddiefang0023@gmail.com (Y.-H.F.); doyeric0926@yahoo.com (H.-Y.C.); ybrpooh719@gmail.com (P.-J.Y.); 2Division of Cardiology, Department of Internal Medicine, National Cheng Kung University Hospital, College of Medicine, National Cheng Kung University, Tainan 70403, Taiwan; 3Department of Physiology, College of Medicine, National Cheng Kung University, Tainan 70101, Taiwan; hhelen000111tw@gmail.com (Z.-H.G.); snwu@mail.ncku.edu.tw (S.-N.W.); 4Institute of Basic Medical Sciences, College of Medicine, National Cheng Kung University, Tainan 70101, Taiwan; 5Center of Cell therapy, National Cheng Kung University Hospital, College of Medicine, National Cheng Kung University, Tainan 70403, Taiwan

**Keywords:** amniotic fluid stem cell, induced pluripotent stem cells, cardiac differentiation, cardiac regeneration

## Abstract

Mature mammalian hearts possess very limited regenerative potential. The irreversible cardiomyocyte loss after heart injury can lead to heart failure and death. Pluripotent stem cells (PSCs) can differentiate into cardiomyocytes for cardiac repair, but there are obstacles to their clinical application. Among these obstacles is their potential for post-transplant rejection. Although human amniotic fluid-derived stem cells (hAFSCs) are immune privileged, they cannot induce cardiac differentiation. Thus, we generated hAFSC-derived induced PSCs (hAFSC-iPSCs) and used a Wnt-modulating differentiation protocol for the cardiac differentiation of hAFSC-iPSCs. In vitro studies using flow cytometry, immunofluorescence staining, and patch-clamp electrophysiological study, were performed to identify the characteristics of hAFSC-iPSC-derived cardiomyocytes (hAFSC-iPSC-CMs). We injected hAFSC-iPSC-CMs intramuscularly into rat infarcted hearts to evaluate the therapeutic potential of hAFSC-iPSC-CM transplantation. At day 21 of differentiation, the hAFSC-iPSC-CMs expressed cardiac-specific marker (cardiac troponin T), presented cardiomyocyte-specific electrophysiological properties, and contracted spontaneously. Importantly, these hAFSC-iPSC-CMs demonstrated low major histocompatibility complex (MHC) class I antigen expression and the absence of MHC class II antigens, indicating their low immunogenicity. The intramyocardial transplantation of hAFSC-iPSC-CMs restored cardiac function, partially remuscularized the injured region, and reduced fibrosis in the rat infarcted hearts. Therefore, hAFSC-iPSCs are potential candidates for the repair of infarcted myocardium.

## 1. Introduction

Human pluripotent stem cells (hPSCs), i.e., human embryonic stem cells (hESCs) and human induced pluripotent stem cells (iPSCs), can differentiate into functionally contracting cardiomyocytes using a well-established Wnt-signaling modulating directed cardiac differentiation protocol [1,2,3,4]. It has been previously shown that hPSC-derived cardiomyocytes (hPSC-CMs) can sufficiently repair damaged cardiac tissues to significantly improve cardiac function [1,2,3,5,6,7]. ENREF_14 Although it seems promising to use hPSC-CMs to repair damaged hearts, there are many hurdles to the clinical application of hPSCs therapy [3,5,8,9]. One of the major obstacles is the potential immune rejection of hPSC-CMs post-transplantation. Currently, the recipients have to rely on taking strong lifelong immunosuppressive agents to prevent rejection of these transplanted cells. These drugs can cause severe adverse events such as renal failure, life-threatening infections, and other malignancies [10,11,12].

Although human amniotic fluid-derived stem cells (hAFSCs) possess the favorable property of pluripotency [13,14,15] and the immune privilege due to low major histocompatibility complex (MHC) class I antigen expression and the absence of MHC class II antigens [13,14,15,16], we demonstrate that hAFSCs cannot differentiate into functional cardiomyocytes in vitro using the Wnt-signaling modulating directed cardiac differentiation protocol [17]. To allow the cardiac differentiation of hAFSCs, we converted hAFSCs to hAFSC-iPSCs. Then we investigated the in vitro characteristics of hAFSC-iPSC-derived cardiomyocytes (hAFSC-iPSC-CMs) and the effect of intramyocardial hAFSC-iPSC-CM transplantation in a rat model with myocardial infarction (MI) to determine whether hAFSC-iPSCs is a good candidate for cardiac regeneration. We showed that hAFSC-iPSC-CMs express cardiac-specific marker, possess cardiomyocyte-specific electrophysiological properties and low immunogenicity and contract spontaneously. Using a relatively less potent immunosuppressive therapy, the engrafted hAFSC-iPSC-CMs could survive in the infarcted area of the rat hearts and restore the post-MI cardiac function. Therefore, hAFSC-iPSCs are potential, ideal candidates for cell therapy.

## 2. Results

### 2.1. Characterization of hPSCs with and without Cardiac Differentiation

hAFSC-iPSCs and hESCs were expanded using α-MEM supplemented with 4 ng/mL of basic fibroblast growth factor (233-FB; R&D Systems, Minneapolis, MN, USA). When these cultured cells reached confluence, we used a serum-free, monolayer direct cardiac differentiation protocol involving a serial application of activin A and bone morphogenetic protein-4 (Figure 1a) [1,2,3,4,16,17] to differentiate both hAFSC-iPSCs and hESCs into cardiomyocytes (hAFSC-iPSC-CMs and hESC-CMs, respectively). At day 21 of cardiac differentiation, we harvested these differentiated cells for characterization studies.

To identify the characteristics of these cells, we first used flow cytometry to show that undifferentiated hAFSCs, hAFSC-iPSCs and hESCs presented the pluripotent stem cell markers, i.e., octamer-binding transcription factor 4 (OCT4), Nanog and stage-specific embryonic antigen-4 (SSEA4), which confirmed the pluripotency of all three types of stem cells tested. In addition, these cells did not express the cardiac-specific marker, cardiac troponin T (cTnT) (Figure 1b). Compared to the undifferentiated stem cells, the three cardiac-differentiating stem cells had minimal expression of the pluripotent stem cell markers but had markedly higher levels of cTnT expression. cTnT levels between hAFSC-iPSC-CMs and hESC-CMs were not significantly different but were notably lower in differentiated hAFSCs (Figure 1b).

Furthermore, it was shown that hAFSCs may have immune privilege due to their close association with maternal tissues in gestation and lack of MHC II antigen expression [16,18]. Both hAFSCs and hAFSC-iPSCs expressed MHC class I antigens (i.e., human leukocyte antigens (HLA)-A, B and C), but did not display MHC class II antigens (such as HLA-DR). In contrast, hESCs represented both MHC class I and II antigens (Figure 1b). It is worth mentioning that, after cardiac differentiation, hAFSC-iPSC-CMs still lack HLA-DR antigen presentation and may thus still be immune privileged (Figure 1b).

Using immunofluorescence staining, we show that both hAFSC-iPSC-CMs and hESC-CMs had apparent cTnT expression (Figure 1c). The immunofluorescence staining results were in concordance with the flow cytometry findings. Moreover, both hAFSC-iPSC-CMs and hESC-CMs, but not cardiac-differentiating hAFSCs, contracted spontaneously. Our results indicated that like hESC-CMs, the hAFSC-iPSC-CMs are true cardiomyocytes. Additionally, we identified connexin 43 gap junction channels in both hAFSC-iPSC-CMs and hESC-CMs (Figure 1d) which indicates the presence of possible electrical communication between the hAFSC-iPSC-CMs [3].

### 2.2. Contractility of hPSC-CMs

At days 14, 21, 30, and 50 of cardiac differentiation, we acquired the images of beating hAFSC-iPSC-CMs and hESC-CMs for analysis. We used the automated non-profit software, MUSCLEMOTION, to quantitate the changes of cardiomyocytes in terms of contractility and relaxation [19]. It was found that there were no significant differences in contraction duration, velocity, and amplitude between hAFSC-iPSC-CMs and hESC-CMs (Figure 2). These results indicated that there was no significant difference in contractility between hAFSC-iPSC-CMs and hESC-CMs. Moreover, the relaxation velocity of hAFSC-iPSC-CMs and hESC-CMs did not differ either. In short, there was no observable difference between the contractile performance of hAFSC-iPSC-CMs and of hESC-CMs.

### 2.3. Characterization of the electrophysiological properties of hAFSC-iPSC-CMs and hESC-CMs

Like hESC-CMs, hAFSC-iPSC-CMs had connexin 43 gap junction channels and could contract spontaneously. To confirm if hAFSC-iPSC-CMs are real cardiomyocytes, we did whole-cell patch-clamp recordings to evaluate the electrophysiological properties of hAFSC-iPSC-CMs and hESC-CMs. At day 7 of cardiac differentiation, when the differentiating cells did not have a spontaneous contraction, the activities of Na^+^ current (*I*_Na_) and L-type Ca^2+^ current (*I*_Ca,L_) of hAFSC-iPSC-CMs and hESC-CMs were limited. However, at day 21 of cardiac differentiation, these cells contracted spontaneously, and the electrophysiological study demonstrated good *I*_Na_ and *I*_Ca,L_ activities (Figure 3A,B). The resting membrane potentials of hAFSC-iPSC-CMs and hESC-CMs were approximately –60~–65 mV.

### 2.4. Therapeutic Effects of hAFSC-iPSC-CMs on Post-Myocardial Infarction (MI) Left Ventricular (LV) Function

To determine the potential therapeutic effects of hAFSC-iPSC-CM transplantation, we used a rat MI model in which MI was induced by an ischemia-reperfusion (I-R) surgery to test whether the intramyocardial transplantation of hAFSC-iPSC-CMs could restore post-MI cardiac function (Table 1 and Figure 4a). Echocardiographic studies were performed on all the animals to assess their LV function one day prior to I-R surgery and two days after MI. At four days post MI, two intramyocardial injections of hAFSC-iPSC-CMs (1×10^7^ cells/kg in a total volume of 80 µL [i.e., ~2.0×10^6^ cells in a volume of 40 µL per injection], *n* = 6) or vehicle (40 µL per injection, *n* = 6, as control) into the infarcted area and the infarct border zone (defined as day 0) were performed (Figure 4a). The final cardiac function was evaluated by echocardiography at four weeks post-infarct (i.e., three weeks after intramyocardial injection, prior to euthanasia) (Figure 4a). Additionally, we also did sham surgery (*n* = 4). Prior to I-R surgery, there was no significant difference in LV systolic function among the three groups, i.e., the sham group, the control group, and the hAFSC-iPSC-CM group. LV ejection fraction (LVEF) and LV fractional shortening (LVFS) were 70.9 ± 4.0% and 40.7 ± 3.7% in the sham group, respectively; 67.2 ± 1.4% and 33.9 ± 3.3% in the control group, respectively; 74.7 ± 2.0% and 35.8 ± 3.4% in the hAFSC-iPSC-CM group, respectively (Figure 4b and 4c). At two days post MI, both the control and the hAFSC-iPSC-CM groups demonstrated significantly decreased LV systolic function (LVEF, from 67.2 ± 1.4% to 48.6 ± 1.9%, *p* = 0.004; FS, from 33.9 ± 3.3% to 15.8 ± 3.4%, *p* = 0.002 in control animals; LVEF, from 74.7 ± 2.0% to 56.2 ± 1.1%, *p* = 0.001; LVFS, from 35.8 ± 3.4% to 24.1 ± 1.4%, *p* = 0.002 in hAFSC-iPSC-CM-treated animals, Figure 4b,c, Table 1).

Furthermore, we tested whether the hAFSC-iPSC-CMs possessed immune privilege *in vivo* compared to the previous study [1,2]. We did not use an immunosuppressive regimen combination of methylprednisolone (2 mg/kg/day, intraperitoneal injection) and cyclosporine A (CsA, 15 mg/kg/day, subcutaneous injection) but chose a slightly lower dose of immunosuppressant with CsA at a dose of 10 mg/kg/day but without methylprednisolone from day –1 to day 21 (euthanasia day). At 3 weeks after treatment, compared to the control group, hAFSC-iPSC-CMs transplantation could significantly improve LV systolic function (LVEF, from 48.6 ± 1.9% to 51.0 ± 0.7%, *p* = 0.35; FS, from 15.8 ± 3.4% to 17.8 ± 2.5%, *p* = 0.66 in control animals; LVEF, from 56.2 ± 1.1% to 65.6 ± 1.1%, *p* = 0.002; FS, from 24.1 ± 1.4% to 34.4 ± 1.4%, *p* = 0.006 in hAFSC-iPSC-CM-treated animals, Figure 4b,c). Additionally, the therapeutic benefit of hAFSC-iPSC-CM intramyocardial transplantation was shown by the change in LVEF and FS between day –2 and day 21. There was a significant improvement of LV systolic function: ∆LVEF 2.4 ± 1.0% and 9.5 ± 0.6% (*p* = 0.003, Figure 4d); ∆LVFS 2.0 ± 1.0% and 10.3 ± 0.6% (*p* = 0.002, Figure 4e), in the control group and the hAFSC-iPSC-CM group, respectively.

Histological studies showed that all the infarcted animals had transmural scarring at four weeks post-MI (Figure 4f). Importantly, despite the relatively less potent immunosuppressive therapy prescribed in our study, those hAFSC-iPSC-CM recipients had partial remuscularization with considerable graft size in the infarcted area (Figure 4g). We used a human-specific HLA-ABC primary antibody which does not cross-reacted with rat specimens, to confirm the human origin of these engrafted cardiomyocytes (Figure 4g). Moreover, there were no teratomas identified in those hAFSC-iPSC-CM-treated hearts.

At three weeks post-transplantation, the gap junction protein connexin 43 was identified in the hAFSC-iPSC-CM grafts and at the intercalated discs between graft and host tissue (Figure 5a). This may imply electrical coupling within the graft tissue and between the hAFSC-iPSC-CM graft and host myocardium. Furthermore, cardiomyocyte proliferation and apoptosis at euthanasia (i.e., three weeks post-transplantation) were measured by staining for Ki67 (Figure 5b) and Annexin V (Figure 5c), respectively. Compared to the control group, more cell cycle activity occurred in the transplanted hAFSC-iPSC-CM grafts, indicating graft cell proliferation (Figure 5b). However, there was no significant difference of cell apoptosis between the vehicle-treated group (control) and the hAFSC-iPSC-CM-grafted group (Figure 5c).

Few CD3^+^ T lymphocytes, CD20^+^ B lymphocytes and CD68^+^ macrophages are found within and around the infarcted region with or without hAFSC-iPSC-CM transplantation. There was no significant difference of CD3^+^ T lymphocytes or CD68^+^ macrophages between the vehicle-treated group (control) and the hAFSC-iPSC-CM-treated group (Figure 6). Nevertheless, compared to the control group, fewer CD20^+^ B lymphocytes were seen in the hAFSC-iPSC-CM-treated group (Figure 6). Our data might indicate that hAFSC-iPSC-CM transplantation might not induce severe immune reaction so that less potent immunosuppressant administration could prevent graft rejection after hAFSC-iPSC-CM intramyocardial transplantation.

In short, our data demonstrated the therapeutic effect of hAFSC-iPSC-CM intramyocardial transplantation in the infarcted region with significant improvement of post-MI LV systolic function in rats with a lower dose of immunosuppressant. It is worth mentioning that the engrafted hAFSC-iPSC-CMs could survive in rat myocardium, even with the lower dose of immunosuppressant, which may imply the potential immune privilege of hAFSC-iPSC-CMs.

## 3. Discussion

In this study, we demonstrated that the intramyocardial transplantation of hAFSC-iPSC-CMs improved post-MI cardiac function. The use of hAFSC-iPSC-CMs might also confer the added advantage of immune privilege and can aid in the prevention of post-transplant rejection. Using a well-established monolayer direct cardiac differentiation protocol, we differentiated hAFSC-iPSCs to contracting cardiomyocytes. Connexin 43 was identified in the cell junctions between hAFSC-iPSC-CMs. Although hAFSC-iPSC-CMs expressed MHC class I molecules, they lacked MHC class II molecules. This hints at a possible immune privilege. Using a relatively less potent immunosuppressive therapy, the engrafted hAFSC-iPSC-CMs could survive in the infarcted area of the rat hearts and restore the post-MI cardiac function.

There is no doubt that PSCs, i.e., ESCs and iPSCs, are able to be differentiated into a large number of beating cardiomyocytes and that PSC-CM transplantation can remuscularize the infarcted hearts and restore post-MI LV function [3,5,20,21]. However, there are several hurdles that have to be overcome prior to the clinical application of cell therapy for cardiac regeneration. One of these major obstacles is the post-transplant rejection of allogeneic cell transplantation [22,23,24]. Owing to the mismatch of HLA antigens, transplanted cells can be rejected. We showed that a combination of several strong immunosuppressive agents is mandatory to prevent the rejection of mismatched hPSC-CM grafts [5]. Nevertheless, the life-long administration of potent immunosuppressants is often accompanied by serious adverse effects such as nephrotoxicity, malignancy and serious/fatal infections [5]. Thus, the administration of strong immunosuppressants is not clinically practical.

A solution to this immunological barrier is to use autologous iPSCs. Although this approach can ensure histocompatibility and reduce the risk of post-transplant rejection, autologous iPSC-CMs transplantation is currently clinically unrealistic due to its high cost and the time required to prepare patient-specific iPSC lines [25,26]. To obtain sufficient numbers of autologous iPSC-CMs for cardiac regeneration, it would usually take at least six months. As a result, autologous iPSC-CM therapy is not ready for use in cardiac regeneration in patients with acute or sub-acute MI. Additionally, genetic and epigenetic variations, which may occur during reprogramming, can influence the behavior of individual clones and lead to immune rejection [27,28].

It was suggested that hAFSCs may have immune privilege due to their close association with maternal tissues in gestation. Although hAFSCs exhibit MHC class I antigens, they lack MHC class II antigens. This leads to defective CD4 T-cell development and function and the lack of helper T cell-dependent antibody production. This may be one of the major advantages of using hAFSCs for allogenic transplantation because it may allow for a certain degree of mismatching and still not result in post-transplant rejection. However, HLA expression could be different after iPSCs induction and cardiac differentiation. In our study, we thus showed that just like hAFSCs, hAFSC-iPSCs and hAFSC-iPSC-CMs displayed MHC class I antigens and did not express MHC class II antigens. These data imply that hAFSC-iPSC-CMs may not be highly immunogenic.

It was proven that MHC class II-matched bone marrow cells can significantly reduce allogeneic rejection after bone marrow transplantation. Therefore, we performed *in vivo* studies to investigate the therapeutic effects as well as the immune privilege of hAFSC-iPSC-CMs. We injected hAFSC-iPSC-CMs (1×10^7^ cells/kg) four days post-MI into the infarcted myocardium of immunocompetent rats. It is worth mentioning that we did not prescribe a combination of potent immunosuppressive agents [2], i.e., high-dose methylprednisolone and high-dose CsA, but only used relatively low-doses of CsA to prevent post-transplant rejection. Three weeks after transplantation, an absolute improvement of approximately 10% was detected in the LVEF of the hAFSC-iPSC-CM-treated group. This mechanical benefit could be caused by the significant engrafted hAFSC-iPSC-CM size in the infarcted region, leading to improving contractile performance (Figure 4d). The survival of these xeno-transplanted hAFSC-iPSC-CMs might have resulted from the immune privilege of hAFSC-iPSC-CM which could allow the use of less potent immunosuppressive therapy for the prevention of post-transplant rejection. 

Furthermore, it is shown that hPSC-derived cardiac progenitor cells and minimally engrafted hPSC-CMs could improve post-MI heart function of the rodents and the non-human primates [29,30]. These studies demonstrated that paracrine effects of human PSC derivatives could also contribute to the amelioration of post-MI heart function. Thus, in our study, the paracrine effect of the transplanted hAFSC-iPSC-CMs could also play a role in enhancing systolic function of the infarcted rat hearts.

We recognize that there were several limitations to this study. First, the group sizes and numbers of our animal studies are limited. We realize that it is mandatory to test more doses of immunosuppressive agents for a thorough investigation and to identify the optimal immunosuppressant dose. Second, it would provide more information if the analyses of proliferation and apoptosis at post-transplant 24 hours and at euthanasia. However, owing to the limited animal numbers in this study, we did not euthanize animals at post-transplant 24 hours so that we could not demonstrate any information of proliferation and apoptosis at post-transplant 24 hours. We acknowledge that such studies should be performed in the future. Third, we did not do *in vivo* experiments to compare the therapeutic effect and the triggered immune response among different types of hPSC-CMs, including hESC-CMs, hAFSC-iPSC-CMs, and hiPSC-CMs (reprogrammed from another source). Although comparing the difference of the therapeutic effect and the immune response among different types of hPSC-CMs is not the goal of this study, it is acknowledged that such studies would provide more information to investigate the triggered immune response of hPSC-CM transplantation and should be performed in the future. Forth, a thorough mechanical investigation of the possible immune privilege of hAFSC-iPSC-CMs is necessary. Last, prior to clinical application, a large animal study is warranted to test the optimal immunosuppressive dose for hAFSC-iPSC-CM transplantation.

## 4. Materials and Methods 

### 4.1. hPSCs Culture and Cardiac Differentiation

hAFSC-iPSCs were generated by iPSC Core Facilities of Human Disease iPSC Service Consortium, Taiwan. Both hAFSC-iPSCs and hESCs (RUES2 cells were used as a positive control for cardiac differentiation; a gift from Dr. Patrick C.H. Hsieh and Dr. Jean Lu, Institute of Biomedical Sciences, Academia Sinica, Taipei, Taiwan) were maintained in α-Minimum Essential Media (MEM) (11900-024; Gibco Invitrogen, Waltham, MA, USA) containing 15% fetal bovine serum (FBS, SH30070.03; HyClone, Boston, MA, USA), 1% glutamine (GlutaMAX Supplement, 35050061; Gibco), and 1% penicillin/streptomycin (15140148; Gibco) under a 5% CO_2_ atmosphere at 37 °C [15]. The protocol was approved by the Institutional Review Boards of National Cheng Kung University Hospital, Tainan, Taiwan (IRB No. A-EX-105-034).

hAFSC-iPSCs and hESCs were expanded using α-MEM supplemented with 4 ng/mL basic fibroblast growth factor (bFGF, AF-100-18B; PeproTech Asia, Rehovot, Israel). Using the serum-free, monolayer direct cardiac differentiation protocol involving a serial application of activin A and bone morphogenetic protein-4 (BMP4) (Figure 1a) [1,2,3,4,16,17], we differentiated both hAFSC-iPSCs and hESCs into cardiomyocytes. All the culture media and the supplemental reagents are listed in Appendix A.

### 4.2. Characterization of Differentiated Cardiomyocytes

#### 4.2.1. Immunofluorescent Staining

At day 21 of cardiac differentiation, hAFSC-iPSC-CMs and hESC-CMs were harvested for characterization by immunofluorescence staining and flow cytometry. These cardiomyocytes were fixed in 4% paraformaldehyde and then washed with phosphate-buffered saline for immunofluorescence staining. The cells were treated with 2% bovine serum albumin (0332; VWR life science, USA) for 1 h at room temperature (22–26 °C) and before incubation with primary antibodies against HLA-ABC, connexin 43, cTnT and α-actinin (Appendix A) overnight at 4°C. After washing with PBS, the samples were incubated with secondary antibodies (Goat anti-mouse Alexa Fluor 488 and Goat anti-rabbit Alexa Fluor 568, Appendix A). The acquired confocal images (BX51; OLYMPUS, Tokyo, Japan) were analyzed and quantified by using Adobe Photoshop.

#### 4.2.2. Flow Cytometry

At day 0 and day 21 of cardiac differentiation, the dissociated cells were stained for cell surface markers including OCT4, Nanog, SSEA4, HLA-ABC and HLA-DR (Appendix A). Cells were permeabilized using permeabilization buffer (554723; BD Biosciences, Franklin Lakes, NJ, USA) prior to intracellular staining for cTnT. Fluorescence characterization and analyses were performed using a BD FACS Canto II (BD Biosciences, Franklin Lakes, NJ, USA).

#### 4.2.3. Contractility Measurements of hAFSC-iPSC-CMs and hESC-CMs

At days 15, 21, 30, and 50 of cardiac differentiation, we used a camera steadily mounted on an inverted microscope with slow-motion features (120 frames/second) to record a video of the beating hAFSC-iPSC-CMs and hESC-CMs for offline analysis. All the acquired files were analyzed using an automated nonprofit software, MUSCLEMOTION [19]. We used MUSCLEMOTION to read and convert the videos to the uncompressed AVI files, and measure the contractility profiles (i.e., contraction amplitude and velocity) of the hAFSC-iPSC-CMs and hESC-CMs [19]. The reference point of relative cell relaxation was obtained according to the change of color in each frame. The noise signal of the interest areas was automatically reduced by MUSCLEMOTION. This software automatically detected the reference frame and measured the contraction profiles of hAFSC-iPSC-CMs and hESC-CMs.

### 4.3. Electrophysiological Measurements

We placed an aliquot of the cell suspension, dissociated by a 1% trypsin/EDTA solution, on a recording chamber mounted on an inverted fluorescence microscope stage (CKX-41; Olympus, Yuanyu, Taipei, Taiwan). The microscope was placed on an anti-vibration air table to avoid mechanical noise and was equipped with a digital video system (DCR-TRV30; Sony, Tokyo, Japan) with 1500× and 40× objective lens. We also used an RK-400 (Bio-Logic, Claix, France) or Axopatch 200B (Molecular Devices, Sunnyvale, CA, USA) amplifier for patch-clamp recording [30]. The cells were bathed in normal Tyrode’s solution at room temperature for patch-clamp experiments. Current and potential signals were monitored using a digital oscilloscope (model 1602; PChome eBay Co., Ltd., Taipei, Taiwan).

The cell-attached clamp recording system was used for the electrophysiological study of hAFSC-iPSC-CMs and hESC-CMs [31,32,33,34]. Data were collected and stored online at 10 kHz using an acquisition interface (Digidata-1440A; Molecular Devices, Sunnyvale, CA), and subsequently analyzed using either pCLAMP 10.7 (Molecular Devices) or 64-bit OriginPro 2016 (Microcal; Scientific Formosa, Inc., Kaohsiung, Taiwan). Through digital-to-analog conversion by Digidata-1440A device, the averaged relationships of voltage-gated *I*_Na_ and voltage-gated *I*_Ca,L_ density versus membrane potential were generated by pCLAMP.

### 4.4. Electrophysiological Data Analysis

To examine the steady-state activation or inactivation kinetics of the *I*_Na_ in hAFSC-iPSC-CMs and hESC-CMs, the current–voltage relationships of this current were least-squares fitted to a Boltzmann function as follows:

$$\frac{I}{I_{max}}=\frac{1}{1+\exp\left[\frac{\pm (V-V_{1/2}}{k}\right]}$$
where *I* is the conductance (i.e., *I*/(V-*V*_rev_)) or membrane current *I*_max_ the maximally activated conductance or current of *I*_Na_, *V* the membrane potential in mV, *V*_1/2_ the membrane potential for a half-maximal activation of the current, and *k* the slope factor of the current (i.e., *I*_Na_ and *I*_Ca,L_).

### 4.5. Rat Myocardial Infarction Model

All the animal experiments were approved by the National Cheng Kung University Animal Care and Use Committee and were conducted in accordance with the Guide for the Care and Use of Laboratory Animals. Sprague Dawley (SD) rats (350~400 g, both genders, approximately 12-weeks old) were anesthetized with a 1 mg/kg Zoletil 50 (Virbac, France) intraperitoneal injection, intubated and mechanically ventilated with a ventilator (mode 683; Harvard, USA), at a frequency of 80 times/minute and a volume of 200 µL, to maintain the adequate respiration of the rat. We performed a thoracotomy to expose the hearts and performed I-R surgery to induce myocardial infarction by completely blocking the mid-left anterior descending coronary artery (LAD) blood flow for 60 min with temporary ligation with 6-0 silk suture. After reperfusion LAD blood flow, we used 4-0 Prolene suture to close the thoracotomy wound and these animals placed in a warm environment to maintain their core temperature at 35~37 °C until recovery from anesthesia. The animals were monitored closely.

### 4.6. Cells Preparation and Intramyocardial Transplantation

The hAFSC-iPSC-CMs at day 21 of cardiac differentiation were used for intramyocardial transplantation. These hAFSC-iPSC-CMs were thawed from a liquid nitrogen tank and suspended in a pro-survival cocktail medium (growth factor-reduced Matrigel (50%; BD; 354230); 100 uM ZVAD (62761; Calbiochem; Massachusetts, USA); 50 nM Bcl-XL BH4 (197217; Calbiochem); 200 nM cyclosporine A (PHR1092; Sigma, Missouri, USA); 100 ng/mL insulin-like growth factor1 (590904; Biolegemd, California, USA); 50 uM pinacidil (P154; Sigma)) [1,2,3]. The cells were kept on ice until injection.

We performed thoracotomy followed by intramyocardial injection 4 days after I-R surgery (defined as day 0). We used a 0.3 mL insulin syringe and a 31-gauge needle for the intramyocardial transplantation of hAFSC-iPSC-CMs (1.0 × 10^7^ cells/kg in a volume of 80 µL pro-survival cocktail medium, *n* = 6) or vehicle (as control, *n* = 6). Moreover, there were 4 animals receiving thoracotomy but without I-R surgery and intramyocardial transplantation (as sham). The rats were anesthetized with 1 mg/kg of Zoletil 50 (Virbac, France), intubated and mechanically ventilated with a ventilator (mode 683; Harvard, USA), at a frequency of 70 times/minute and a volume of 200 µL. All animals received two direct intramyocardial injections (~2 × 10^6^ cells in a volume of 40 µL per injection) performed according to the following protocol: one injection at the center of the infarcted area and one injection at the border zone of the infarcted area. The chest walls were closed with 4-0 Prolene. Thereafter, the painkillers and antibiotics were administered to reduce the post-surgical pain and the risks of infection. All rats received a subcutaneous injection of CsA (10 mg/kg/day), starting one day prior to engraftment (day −1) and continuing until euthanasia (3 weeks after intramyocardial injection) [1].

### 4.7. Cardiac Function Evaluation

Cardiac function was evaluated by repeated echocardiography 1 day prior to I-R surgery, 2 days and then 4 weeks post-infarct (i.e., 3 weeks after intramyocardial injection). The animals were lightly anesthetized with inhaled isoflurane (Novaplus) and scanned by transthoracic echocardiography (vevo 770; VisualSonics, Toronto, Canada) with a 10 MHz probe. LV end-diastolic dimension (LVEDD), LV end-diastolic volume (LVEDV), LV end-systolic dimension (LVESD), and LV end-systolic volume (LVESV) were acquired to measure LVEF and FS. LVEF and FS were calculated using the following equations: LVEF = [(LVEDV-LVESV)/LVEDV] × 100(%); FS = [(LVEDD-LVESD)/LVEDD] × 100(%). All investigators performing echocardiographic acquisition and analysis were blinded to the animal groups.

### 4.8. Histology and Immunohistochemistry

Rats were euthanized 3 weeks after cells or vehicle transplantation by intravenously injection of phenobarbital and phenytoin (Beuthanasia-D). The hearts were harvested, rinsed in non-sterile saline, and perfused with 0.9% saline and 4% paraformaldehyde sequentially. The right and left atria and the right ventricle were harvested. LV was sliced parallel to the short axis at 1-mm thickness. Picrosirius red/fast green staining was performed to determine the infarct/fibrotic area. Infarct size was calculated as Σ (infarct area/block area) and expressed as a percentage of the left ventricle. The hAFSC-iPSC-CM graft was identified by immunofluorescence staining with primary antibodies directed against α-actinin and human-specific HLA-ABC (Appendix A) and the fluorescent secondary antibodies (Alexa-conjugated, species-specific antibodies, Appendix A). Graft size was determined as Σ (graft area/infarct area) and expressed as a percentage of the infarct region. 

To check post-transplant immune reaction, we performed immunohistochemistry staining with primary antibodies against CD3, CD20 and CD68 (Appendix A) and EnVision Detection System, Peroxidase/DAB kit (DAKO, K5007), followed by the HRP secondary antibody. Images were acquired using an Olympus BX51 microscope (OLYMPUS, Tokyo, Japan) and quantitated using Image J software. 

### 4.9. Statistical Analyses

Continuous data are expressed as means ± standard error of the mean (S.E.M.). Nonparametric Kruskal–Wallis tests were used for comparisons as the data were not normally distributed. The statistical significance was defined as a two-sided *p*-value of less than 0.05. Statistical analyses were performed using SPSS version 22.0 (IBM Corp., Armonk, NY, USA).

## 5. Conclusions

Just like hESC-CMs, hAFSC-iPSC-CMs express cardiac-specific marker, possess cardiomyocyte-specific electrophysiological properties, and contract spontaneously. Importantly, our in vitro and *in vivo* data indicate that these hAFSC-iPSC-CMs are likely to have low immunogenicity. Taken together, hAFSC-iPSCs are potential, ideal candidates for cell therapy for cardiac regeneration.

## Figures and Tables

**Figure 1 ijms-21-02359-f001:**
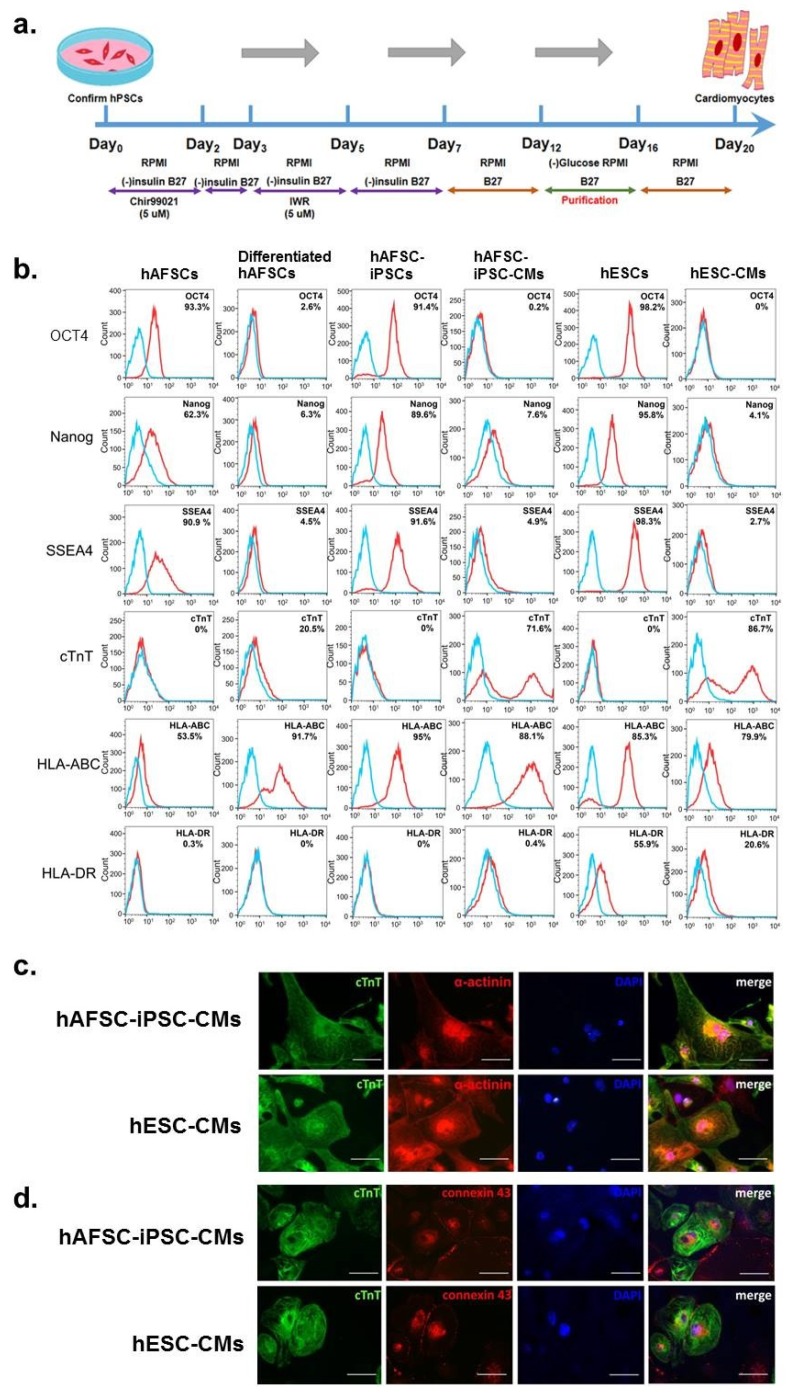
Characterization of human amniotic fluid-derived stem cells (hAFSCs) and hAFSC-induced pluripotent stem cells (hAFSC-iPSCs). (**a**) Schematic presentation of the monolayer direct cardiac differentiation protocol. (**b**) Flow cytometry analyses of cell markers were performed to characterize undifferentiated and cardiac-differentiated stem cells. Pluripotency markers (i.e., OCT4, Nanog and SSEA4), cardiac-specific marker (cardiac troponin T, cTnT), and immune-related surface markers (human leukocyte antigens [HLA]-ABC and HLA-DR) were stained to characterize the hAFSCs and the hAFSC-iPSCs 21 days after cardiac differentiation. Human embryonic stem cells (hESCs) and hESC-derived cardiomyocytes (hESC-CMs) were used as controls. Isotype controls are in blue and the surface markers are in red. The expression of each surface marker or isotype control was analyzed on 100,000 cells. Like hESC-CMs, the hAFSC-iPSC-CMs, but not differentiated AFSCs, had a typical bimodal distribution of cardiac differentiation. Furthermore, both hAFSC-iPSCs and hAFSC-iPSC-CMs expressed HLA-ABC but not represent HLA-DR. (**c**) Immunofluorescence staining revealed that hESC-CMs and hAFSC-iPSC-CMs expressed cTnT at day 21 of cardiac differentiation. (**d**) Connexin43 gap junction channels were identified in both hESC-CMs and hAFSC-iPSC-CMs. Scale bar of c & d, 50 µm.

**Figure 2 ijms-21-02359-f002:**
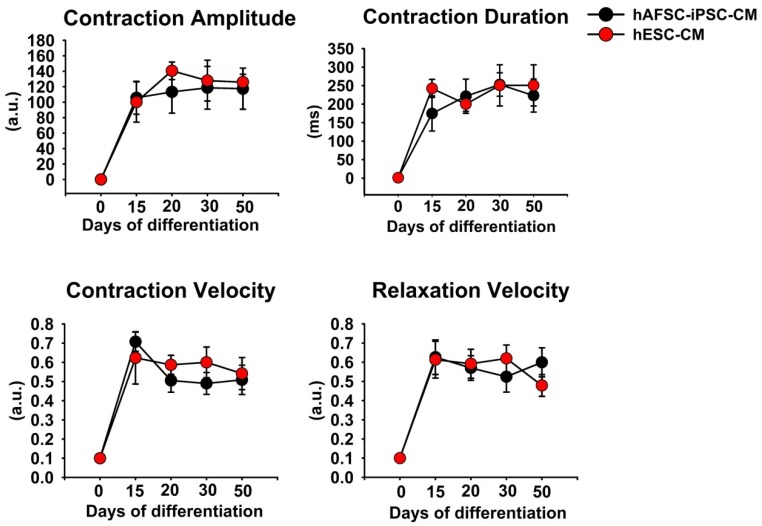
Analyzing in vitro contractile performance of human amniotic fluid-derived stem cells-induced pluripotent stem cell-derived cardiomyocyte (hAFSC-iPSC-CM) and human embryonic stem cell-derived cardiomyocyte (hESC-CM). An automated non-profit software, MUSCLEMOTION, was used to quantitate the in vitro contractile performance. There were no significant differences in contractile function (including contraction amplitude, duration, and velocity) and relaxation of all the data points between these hAFSC-iPSC-CMs and hESC-CMs. Each data point represents the mean ± S.E.M. (*n* = 5).

**Figure 3 ijms-21-02359-f003:**
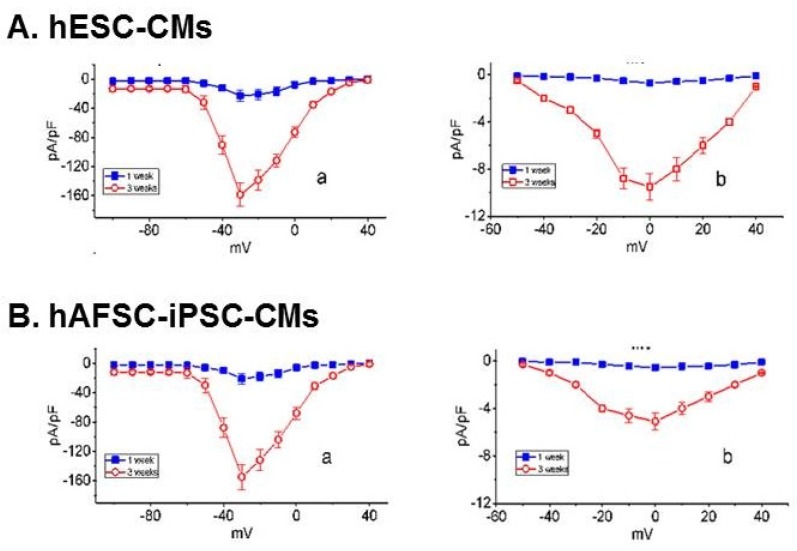
The averaged relationships of voltage-gated Na^+^ current (*I*_Na_, **A**a and **B**a) or voltage-gated L-type Ca^2+^ current (*I*_Ca,L_, **A**b, and **B**b) density versus membrane potential recorded either from hESC-CMs (panel A) or hAFSC-iPSC-CMs (panel B) at 1- and 3-week of cardiac differentiation. To measure *I*_Na_ or *I*_Ca,L_, cells were bathed in Ca^2+^-free Tyrode’s solution while the recording electrode was backfilled with Cs^+^-containing solution. To elicit *I*_Na_, the examined cells were briefly depolarized from –80 to various potentials ranging between –100 and +40 mV in 10-mV step. For the recording of *I*_Ca,L_, the cells were voltage-clamped at –50 mV and the depolarizing command potentials ranging between –50 and +40 mV were applied. Each data point represents the mean ± SEM (*n* = 8).

**Figure 4 ijms-21-02359-f004:**
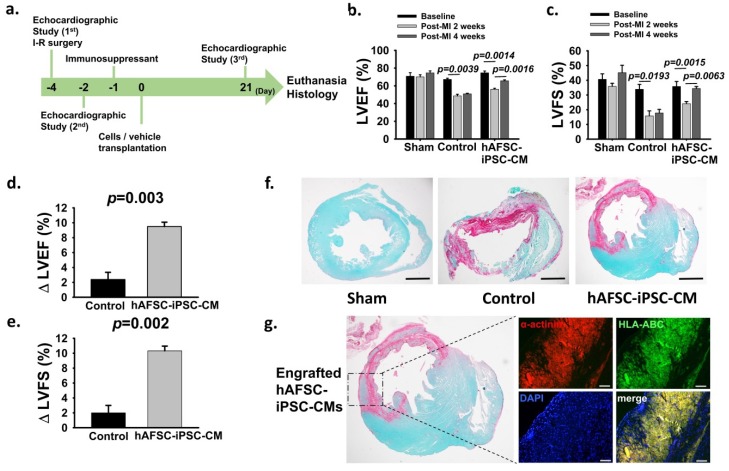
Therapeutic effects of hAFSC-iPSC-CM transplantation. (**a**) Timeline for the efficacy study in which immunocompetent 8-week-old Sprague Dawley (SD) rats (350~400 g, both genders) received 60-minute ischemia, followed by reperfusion in the mid-left anterior descending coronary artery. Echocardiographic studies were performed prior to the infarct surgery 2 days prior to cell transplantation and 3 weeks after transplantation. (**b**,**c**) All the animals (including the control group and the hAFSC-iPSC-CM-treated group) had deteriorating cardiac function after myocardial infarction. Cardiac function studies (b, LVEF; c, LVFS) showed a significant improvement after hAFSC-iPSC-CMs treatment. (**d**,**e**) Changes in LVEF (∆ LVEF, d) and LVFS (∆ FS, e) from 2 days after myocardial infarction to 3 weeks after treatment were significantly greater in hAFSC-iPSC-CM-treated group. (**f**) A representative stain of a short-axis cross-section of SD rat hearts with picrosirius red and fast green. The fibrotic infarct region is red and healthy myocardium is green. There was no fibrotic tissue in the sham group. However, the infarct is transmural in the hearts of the control and the hAFSC-iPSC-CM-treated groups. Scale bar, 2mm. (**g**) hAFSC-iPSC-CM-engrafted heart demonstrating several human myocardium islands of considerable size in the infarcted region. These human myocardia were stained with human specific HLA-ABC antibody. Confocal immunofluorescence scale bar, 50 µm.

**Figure 5 ijms-21-02359-f005:**
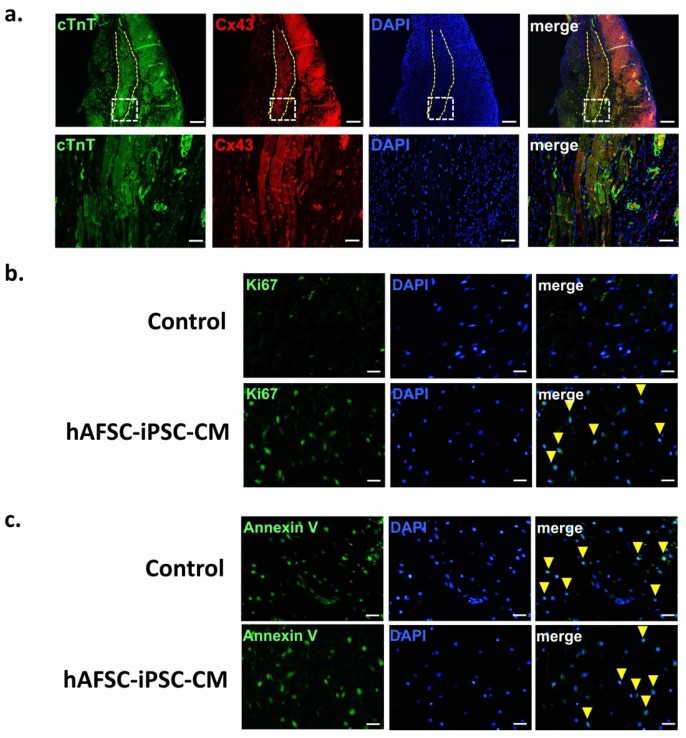
Graft integration, proliferation and apoptosis. (**a**) The infarcted region (dashed line) was remuscularized with hAFSC-iPSC-CM graft. Connexin 43 (Cx43) expression was detected at post-transplant 21 days (upper panel). The transplanted cardiomyocytes at the edges of grafts (lower panel, the magnified image of region boxed in the upper panel) showed Cx43-positive gap junctions between the graft and host. Confocal immunofluorescence scale bar, 50 µm (upper panel) and 200 µm (lower panel). (**b**) Proliferation of 3-week hAFSC-iPSC-CM graft was demonstrated by staining for Ki67 (green). Those cardiomyocyte nuclei in proliferation were indicated by arrowhead. Scale bar, 50 µm. (**c**) Apoptosis at post-transplant 3 weeks was demonstrated by staining for annexin V (green, arrowhead). Scale bar, 50 µm.

**Figure 6 ijms-21-02359-f006:**
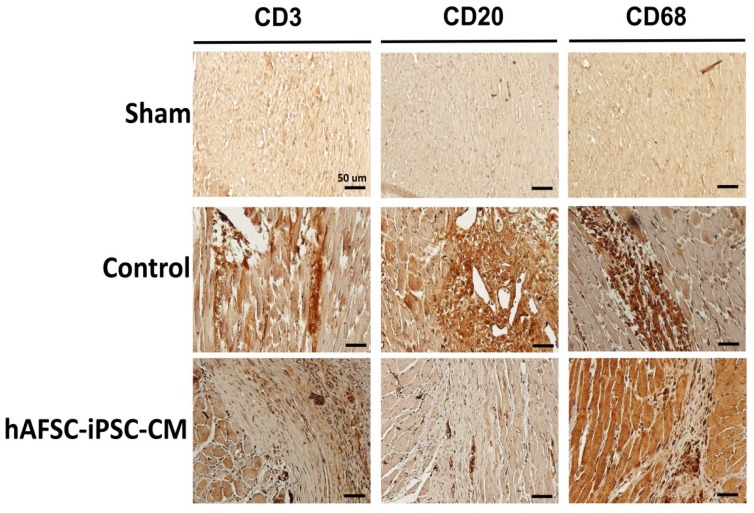
Immune response of the infarcted region. Immunohistochemistry image of rat hearts subjected to myocardial infarction and transplantation of either vehicle (control) or hAFSC-iPSC-CMs at post-transplant 3 weeks. Few CD3^+^ T lymphocytes, CD20^+^ B lymphocytes and CD68^+^ macrophages are seen surrounding the hAFSC-iPSC-CM graft. Comparable numbers of T and B lymphocytes and macrophages were also detected in the infarcted myocardium of the control animals receiving vehicle treatment, but not in the myocardium of the sham animals (i.e., those animals without myocardial infarction). Scale bar, 50 µm.

**Table 1 ijms-21-02359-t001:** Characteristics of morphometry and left ventricular function.

Demographic / Cardiac Parameters		Sham (*n* = 4)	Vehicle-Injected Group (Control; *n* = 6)	hAFSC-iPSC-CM Transplanted Group (*n* = 6)
Age (weeks)		12.0 ± 0.2	12.0 ± 0.5	12.0 ± 0.3
Body weight (g)		379.3 ± 9.8	358.1 ± 6.6	361.3 ± 7.2
LVFS (%)	Pre-MI	40.7 ± 3.7	33.9 ± 3.3	35.8 ± 3.4
	Post-MI 2 days	35.9 ± 2.0	15.8 ± 3.4	24.1 ± 1.4
	Post-MI 4 weeks	45.2 ± 5.0	17.8 ± 2.5	34.4 ± 1.4
LVEF (%)	Pre-MI	70.9 ± 4.0	67.2 ± 1.4	74.7 ± 2.0
	Post-MI 2 days	70.1 ± 2.4	48.6 ± 1.9	56.2 ± 1.1
	Post-MI 4 weeks	74.6 ± 2.3	51.0 ± 0.7	65.6 ± 1.1
∆ LVEF (%), between post-MI 2 days and 4 weeks		N/A	2.4 ± 1.0	9.5 ± 0.6
Infarct size (% LV)		N/A	13.3 ± 4.1	11.2 ± 0.8
Graft size (% infarct)		N/A	N/A	9.1 ± 1.6

Abbreviation: hAFSC-iPSC-CM, human amniotic fluid stem cell derived induced pluripotent stem cell differentiated cardiomyocyte; LVEF, left ventricular ejection fraction; LVFS, left ventricular fractional shortening; MI, myocardial infarction; N/A, not applicable.

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
