# Peer review of "Efficient Cardiac Differentiation of Human Amniotic Fluid-Derived Stem Cells into Induced Pluripotent Stem Cells and Their Potential Immune Privilege"

_ijms, 2020, doi:10.3390/ijms21072359_

Round 1

Reviewer 1 Report

This manuscript dealt with developing a protocol from hAFSC-reprogrammed iPSCs differentiating into hAFSC-iPSC-CM to reduce the risk of transplantation rejection. The article was well written and organized. I only have several suggestions to increase the readability.

In Figure 2, how many samples does each data point represent? Does any of these points show the statistical significance? The authors may think to add statistical symbol in the figure. Also, the legend to indicate hAFSC-iPSC-CM and hESC-CM only shows in the first chart. The authors can try to move it to outsides of these four charts, at the top or right side.

There is no description for Figure 4a in the main text.

The main question for this article is the lack of data showing hESC-CMs and hiPSC-CM (reprogrammed from other sources) in the animal model, to compare the therapeutic effects and the triggered immune response.

Author Response

Responses to reviewers’ comments

Reviewer 1 comments:

1. In Figure 2, how many samples does each data point represent? Does any of these points show the statistical significance? The authors may think to add statistical symbol in the figure. Also, the legend to indicate hAFSC-iPSC-CM and hESC-CM only shows in the first chart. The authors can try to move it to outsides of these four charts, at the top or right side.

Response: Each data point represented 5 samples. All of these points did not show statistical significance. We have mentioned this in the revised figure legend. We also moved the legend which indicated hAFSC-iPSC-CM and hESC-CM to the right side of the charts (Page 5, line 141).

2. There is no description for Figure 4a in the main text.

Response: Thank you. We revised our manuscript and add the description for Figure 2a in the main text (Page 6, line 162-171).

3. The main question for this article is the lack of data showing hESC-CMs and hiPSC-CM (reprogrammed from other sources) in the animal model, to compare the therapeutic effects and the triggered immune response.

Response: Thank you for your important and critical points. We did not have any data to show the difference of therapeutic effects and the triggered immune response among hESC-CMs, hAFSC-iPSC-CMs and hiPSC-CMs (reprogrammed from other sources). This is our major weakness. We recognized this and mentioned this important point in our discussion (study limitations; page 11-12, line 330-336).

  Furthermore, we have 10 days to revise our manuscript. Although we have hESC-CMs and hiPSC-CMs (reprogrammed from other sources) in our lab, we could not complete such a study in time: revising our IACUC, doing I-R surgery, cells transplantation and euthanasia at 3 weeks after cell transplantation.  However, we acknowledge the importance to do the comparison between different cardiomyocytes derived from different pluripotent stem cells. Therefore, we mentioned this in our discussion, too (page 12, line 334-336).

Reviewer 2 Report

The manuscript by Fang et al describes the use of amniotic fluid stem cells to derive HLA-DR-negative induced pluripotent stem cells and their differentiation into functional cardiac myocytes.

The conclusions drawn by the manuscript are very interesting for a potential clinical translation. However, there remain some unclear points.

As first, the number of animals used in the in vivo experiments of the study is not specified clearly. A table with the different groups considered should be included.

The modality for cell injection has not been completely described. Information as multiple or single injection/s, volume per single injection, cell density per single injection, mapping of the ischemic area, the specific site/s of injection is missing.

Reported results on heart performance after cell injection are very promising but should be implemented with an evaluation of cardiac troponin T, as well as connexin 43 to demonstrate cell coupling. 

Analysis of proliferation and apoptosis at 24 hours, as well as at euthanasia should be performed.

Since xenotransplantation has been performed, it would be also essential to demonstrate the absence of any inflammatory and immune response as it is apparent from the other results described in the manuscript.

The authors should also comment in the discussion on whether a paracrine effect could be also considered since many publications on the topic of cell therapy, included with amniotic fluid cells, demonstrated that extracellular matrix vesicles play a major role in the amelioration of heart conditions after acute ischemia.

Author Response

Reviewer 2 Comments and Suggestions for Authors

1. As first, the number of animals used in the in vivo experiments of the study is not specified clearly. A table with the different groups considered should be included.

Response: I am sorry to make you being confused. We mentioned the animal number each group in the revised manuscript and also added a table as your recommendation (page 6, line 168; page 14, line 437-440).

2. The modality for cell injection has not been completely described. Information as multiple or single injection/s, volume per single injection, cell density per single injection, mapping of the ischemic area, the specific site/s of injection is missing.

Response: Thank you for your important point. We described the information of cell transplantation in our revised text (page 14, line 431-440).

3. Reported results on heart performance after cell injection are very promising but should be implemented with an evaluation of cardiac troponin T, as well as connexin 43 to demonstrate cell coupling. 

Response: We did double staining with cardiac troponin T and connexin 43 to show cell coupling (Figure 5a). We mentioned our findings in the main text and figure legend (page 8, line 226-229).

4. Analysis of proliferation and apoptosis at 24 hours, as well as at euthanasia should be performed.

Response: As your important recommendation, we did immunofluorescence staining to show proliferation and apoptosis at euthanasia (3 weeks after transplantation) (Figure 5b and 5c, page 8-9, line 229-234). Because we did not euthanize animals at 24 hours after intramyocardial transplantation, we can not analyze proliferation and apoptosis at 24 hours after cell transplantation. We recognized that this is one of our study limitations so we mentioned this weakness in our discussion (page 11, line 325-329).

I have to mention that we have 10 days to revise our manuscript. Because we have to revise our IACUC which could need 3~4 weeks to complete the IACUC revision, and then do I-R surgery, cell transplantation and euthanize the animal 1 day later, thus we could not complete such a study in time.  However, we acknowledge the importance of this study. Therefore, we mentioned this in our discussion (page 11, line 325-329).

5. Since xenotransplantation has been performed, it would be also essential to demonstrate the absence of any inflammatory and immune response as it is apparent from the other results described in the manuscript.

Response: Thank you for your important suggestion. We did immunohistochemistry staining with CD3 (for T cell), CD20 (B cell) and CD68 (macrophage) to show the immune response (Figure 6). We described our finding in the manuscript (page 9, line 245-252).

6. The authors should also comment in the discussion on whether a paracrine effect could be also considered since many publications on the topic of cell therapy, included with amniotic fluid cells, demonstrated that extracellular matrix vesicles play a major role in the amelioration of heart conditions after acute ischemia.

Response: Thank you. We discussed the role of paracrine effect after cell therapy in our discussion (page 11, line 316-321).

Round 2

Reviewer 2 Report

The authors replied to all the comments of the reviewer.

The reviewer may only suggest modifying the title of the manuscript to better reflect the interesting results achieved and their clinical potential.

The reviewer wishes to express the authors compliments for the results achieved and believes these research outcomes will have a high impact on the clinical translation.